# Perturbation theory without power series:
# iterative construction of non-analytic operator spectra

Matteo Smerlak[1*]

**1** Max Planck Institute for Mathematics in the Sciences, Leipzig, Germany *
CorrespondingAuthor@email.address

February 14, 2022

## Abstract

**It is well known that quantum-mechanical perturbation theory often give rise to divergent series that require proper resummation. Here I discuss simple ways in which these divergences can be avoided in the first place. Using the elementary technique of relaxed fixed-point iteration, I obtain convergent expressions for various challenging ground states wavefunctions, including quartic, sextic and octic anharmonic oscillators, the hydrogenic Zeeman problem, and the Herbst-Simon Hamiltonian (with finite energy but vanishing Rayleigh-Schrödinger coefficients), all at arbitarily strong coupling. These results challenge the notion that non-analytic functions of coupling constants are intrinsically "non-perturbative". A possible application to exact diagonalization is briefly discussed.**

**Introduction.** Dyson was the first to observe that, since quantum electrodynamics with a negative fine structure constant is unstable, the perturbative series in powers of $\alpha$ must diverge in the physical sector with $\alpha > 0$, no matter how small its numerical value [1]. Unfortunately, such asymptotic series with zero radius of convergence are the rule rather than the exception: simple problems such as the one-dimensional quartic anharmonic oscillator or the hydrogen atom in an external field already give rise to perturbative series that diverge for any coupling strength [2]. Similar issues arise in molecular [3] and nuclear [4] physics, quantum chemistry [5], and field theory [6].

The usual strategy to deal with a divergent perturbative series is to attempt to *resum* it, *i.e.* to use its coefficients to construct a convergent expression in ways other than simple summation [7–9]. Various procedures have been developed for this purpose, including sequence transformations [10], Padé approximants [11], Borel-Laplace resummation [12–14], and order-dependent mappings [15]. Applying these procedures in practice can be difficult. For starters, since the perturbative coefficients grow factorially or super-factorially, high-precision (or exact) arithmetics is usually required. Second, methods based on analytic continuations (*e.g.* Borel resummation or order-dependent mappings) require *a priori* knowledge of the singularity structure of the eigenvalues, which must obtained through semiclassical estimates [7] or some other technique [16], and can give rise to ambiguities [14].

Worse still, resummation is not always possible. Herbst and Simon exhibited an anharmonic oscillator with finite ground state energy $E(g) \sim e^{-d/g^2}$ (with $d > 0$) but vanishing perturbative coefficients at all orders [17], meaning that perturbation theory contains *no* information about the ground state. To make progress, Jentschura and Zinn-Justin leveraged semiclassical estimates to conjecture a modified Born-Sommerfeld quantization condition [18,19] which gives the energy as a resurgent expansion [20]. Combined with Borel-Padé resummation, this approach allows to compute $E(g)$ at small coupling values $g$ [21]. However—like other resummation procedures—the method does not by itself allow to construct the corresponding eigenvector.

The basic problem underlying the divergence of perturbative expansions is that a function $E(g)$ of a variable $g$ with a singularity at $g_* \in \mathbb{C}$—e.g. "intruder state" [22]—cannot be expressed as a convergent powers series in $g$ outside a disk of convergence of radius $|g_*|$ (Abel's lemma); when such a singularity lies at $g = 0$, a power series expansions can never converge. Resummation theory is an effort to bypass this obstruction by interpreting the perturbation series as the Taylor expansion of some member of a more general, yet sufficiently rigid, class of function of $g$: a rational function for Padé approximants, a resurgent function for Borel-Ecalle resummation, etc. A common view is that non-analytic contributions to the energy function $E(g)$ (e.g. instantons $\sim e^{-d/g^2}$), being intrinsically "non-perturbative", can only be captured from the large order behavior of perturbation theory, if at all.

But there is more to perturbation theory than Taylor expansions in the theory's coupling constant $g$. First, given a Hamiltonian $H = F + gI$ (where $F$ stands for "free" and $I$ for "interaction"), nothing dictates that we treat $F$ as the unperturbed Hamiltonian and $gI$ as the perturbation. When we use any another partitioning into $H = H_0 + \lambda H_1$ (with $H_0$ a spectral function of $F$, with the same eigenvectors), the Rayleigh-Schrödinger (RS) algorithm yields a solution which is a power series in $\lambda$, but not in $g$. Second, whether we use $F$ or some other $H_0$ as unperturbed Hamiltonian, assuming a power series ansatz is not necessarily the most straightforward approach to solving the perturbation equation. I discuss below an alternative formulation of perturbation theory, called "iterative perturbation theory" (IPT), that does not rely on this ansatz and is more computationally efficient.

These observations are elementary, but also empowering. In the following I show that convergent approximations of the ground state energy—and eigenvector—of even anharmonic oscillators, the Zeeman problem and the Herbst-Simon Hamiltonian can be constructed perturbatively, at any coupling strength and without the need for educated guesses, extraneous semiclassical estimates, ingenious resummation techniques, or knowledge of the nature and location of singular points in the complex $g$ plane.

Many authors have noted that the convergence of perturbation theory can be improved by using the freedom to repartition the Hamiltonian [15, 23–28]. However, the large-order behavior of these optimized schemes has rarely been analyzed. To my knowledge, the resummation-free construction of a non-trivial quantum-mechanical ground state at arbitrarily high coupling strength with these methods has not been reported (with the exception of a zero-dimensional model [29]). More importantly—as I hope to convince the reader—none of these optimized perturbation methods is as elementary as the relaxed iterative scheme proposed here.

**Eigenvalue perturbation theory.** Consider a Hamiltonian of the form $H = H_0 + \lambda H_1$ where $H_0$ has a known spectrum and $H_1$ is a perturbation. (The method also applies to non-Hermitian problems, but we restrict to Hamiltonian operators here for simplicity). Here $\lambda$ is a perturbation parameter which we will eventually set to 1. We focus on an isolated, non-degenerate, stable[1] eigenvalue $E_0$ with corresponding eigenvector $\psi_0$ and eigenprojection $P_0$. We seek an eigenvector $\psi$ of $H$ with eigenvalue $E$ and $\langle \psi_0 | \psi \rangle = 1$. Under this normalization condition, the eigenvalue equation $H\psi = E\psi$ is equivalent to

$$\psi = \psi_0 + \lambda R_0(\mathcal{E})(H_1 + \mathcal{E} - E)\psi,$$

where $R_0(\mathcal{E}) = (I - P_0)(\mathcal{E} - H_0)^{-1}(I - P_0)$ is the reduced resolvent. In the following we focus on the choice $\mathcal{E} = E_0 = E - \langle \psi_0 | H_1 \psi \rangle$, which turns the perturbation equation into the quadratic equation[2]

$$\psi = \psi_0 + \lambda R_0(E_0)(H_1 - \langle \psi_0 | H_1 \psi \rangle). \tag{1}$$

---

[1]Stability here means that the eigenvalues remains isolated and non-degenerate for small values of $\lambda$.

[2]Choosing instead $\mathcal{E} = E$ is the starting point of Brillouin-Wigner perturbation theory, in which $E$ is computed implicitly rather than explicitly.

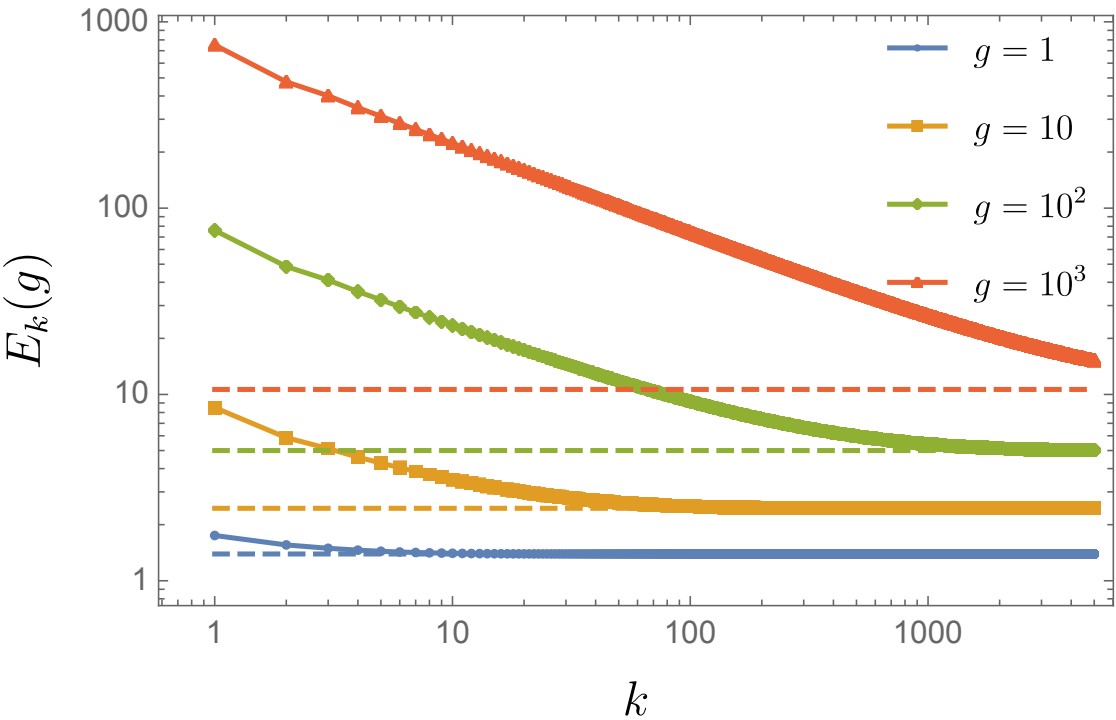

Figure 1: Ground state energy of the quartic oscillator computed with IPT ($\alpha = 1/2$) under EN partitioning (6) (continuous lines) compared to exact diagonalization in a basis set with $5 \cdot 10^4$ harmonic oscillator states (dashed lines), for increasingly large coupling constants $g$. Eigenvectors and eigenvalues both converge (albeit slowly) without the need for any resummation procedure.

In a broad sense, perturbation theory consists in finding solution to (1) iteratively. There is more than one way to approach this problem.

**RS perturbation theory.** Conventional RS theory seeks solutions of (1) as a power series $\psi_{\text{RS}} = \sum_{l \geq 0} \lambda^l a_l$. Inserting this ansatz into (1) and collecting terms of the same order in $\lambda$ gives the well-known recursion relation $a_{l+1} = Q_{\text{RS}}(a_0, \cdots, a_l)$ with

$$Q_{\text{RS}}(a_0, \cdots, a_l) \equiv \lambda R_0(E_0)[H_1 a_l - \sum_{s=0}^{l} \langle \psi_0 | H_1 a_s \rangle a_{l-s}] \tag{2}$$

with initial condition $a_0 = \psi_0$. In many cases of physical interest (including all examples below, and others [30]), the perturbation $H_1$ is block-diagonal in the basis of unperturbed eigenstates, hence (2) is effectively a final-dimensional problem which can be solved at any finite order $k$ using computer algebra[3]. (When this condition is not met, the perturbation equations must usually be solved numerically in a finite basis set, *i.e.* perturbation theory becomes a variational approximation.) From these eigenvector coefficients the energy can be computed at any order $k$, either directly as $E_{\text{RS}}^{(k)} = \sum_{l \geq 0}^{k-1} \lambda^l \langle \psi_0 | H a_l \rangle$, or more efficiently through Wigner's $2n + 1$ theorem.

**Iterative perturbation theory.** An alternative approach, termed here "iterative perturbation theory" (IPT), starts from the observation that (1) is a fixed-point equation $\psi = F(\psi)$ for the

---

[3]In particular, to any finite perturbative order $k$, all sums over unperturbed eigenstates are all finite sums.

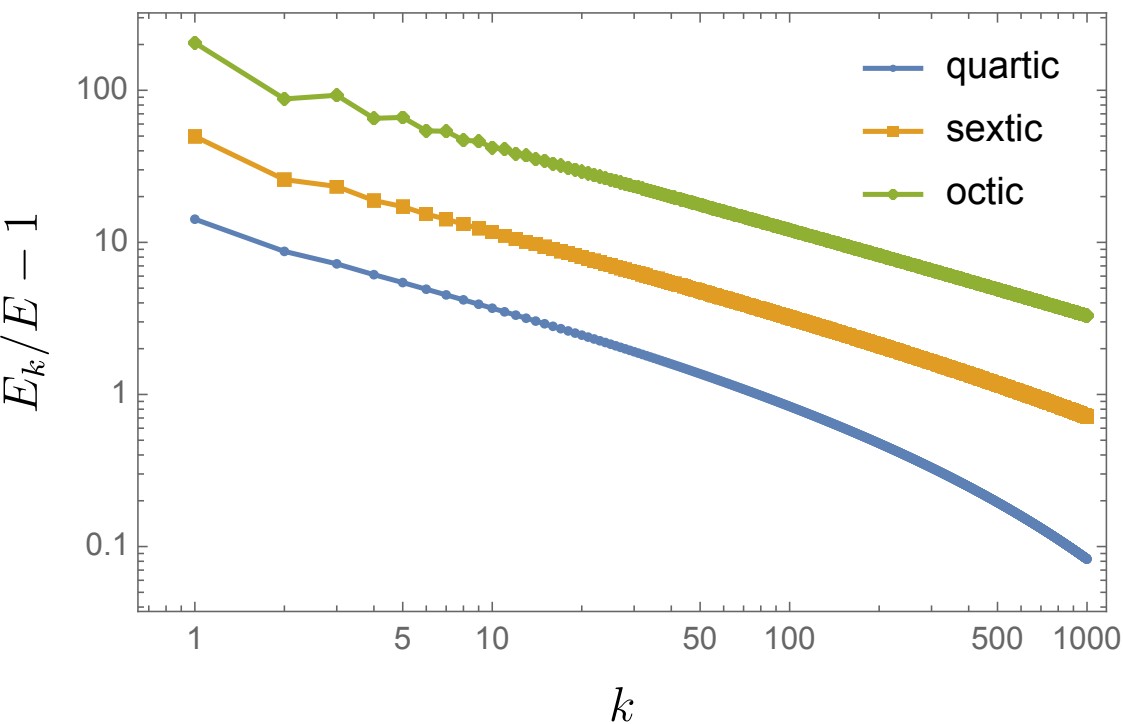

Figure 2: Convergence of IPT energy iterates $E^{(k)}$ for the ground states of the quartic, sextic and octic anharmonic oscillators, all with $g = 100$. Here we used $\alpha = 1/2$ and EN partitioning; $E$ is obtained by exact diagonalization. Note the doubly-logarithmic axes: convergence is slow for these examples.

non-linear operator

$$Q_{\mathrm{IPT}}(\psi) \equiv \psi_0 + \lambda R_0(E_0)(H_1 - \langle \psi_0 | H_1 \psi \rangle). \tag{3}$$

This suggests that $\psi$ can also be computed dynamically, as the limit of the sequence of iterates $\psi_{\mathrm{IPT}}^{(k+1)} = Q_{\mathrm{IPT}}(\psi^{(k)})$ starting from $\psi^{(0)} = \psi_0$. The corresponding energy given by $E_{\mathrm{IPT}}^{(k)} = \langle \psi_0 | H \psi_{\mathrm{IPT}}^{(k-1)} \rangle$. This approach is *a priori* more efficient than the RS recursion (2): IPT is a first-order recurrence relation, whereas (2) involves all previous $a_l$ coefficients. Moreover, IPT makes it easy to prove convergence results using Banach's fixed point theorem. For instance, we can show that, if $H_1$ is bounded, $\psi^{(k)}$ converges as $k \to \infty$ for any $|\lambda| < (3 - 2\sqrt{2})/\|H_1\|\delta$ where $\delta$ the distance between $E_0$ and the rest of $H_0$'s spectrum [31]. The corresponding result in RS theory is "not at all trivial" *dixit* Kato [32]. Finally, IPT is directly related to RS as follows: after $k$ iterations, $\psi_{\mathrm{IPT}}^{(k)}$ contains powers of $\lambda$ with exponents up to $2^k - 1$, but low-order terms coincide with RS, *viz.* $\psi_{\mathrm{IPT}}^{(k)} = \psi_{\mathrm{RS}}^{(k)} + \mathcal{O}(\lambda^{k+1})$, as can be easily seen by recursion.

**Relaxation.** Both RS and IPT involve recurrence relations of the form $X_{n+1} = Q(X_n)$. The main purpose of this paper is to direct attention to the fact that the convergence of perturbative calculations is much improved if we use instead the *relaxed* iteration procedure

$$X_{n+1} = \alpha Q(X_n) + (1 - \alpha)X_n \text{ for some } 0 < \alpha < 1. \tag{4}$$

Relaxation is a well-known technique in numerical analysis: it simply amounts to choosing a smaller time step in the Euler discretization of a dynamical system—a natural strategy to improve convergence. From the perspective of perturbation theory, (4) turns out to be equivalent

to a repartitioning of the Hamiltonian into

$$H = [\alpha^{-1}H_0] + [H_1 + (1 - \alpha^{-1})H_0]. \tag{5}$$

Eq. (5) with a special choise of $\alpha$ is known in quantum chemistry as "Feenberg scaling" [33–35]. Such a repartitioning is also the starting point of order-dependent mappings [15] and of self-consistent expansions [25]. They key point is that, upon such a repartitioning, neither RS nor IPT give rise to power series of $\lambda^4$, but instead to expansions of the form $\psi^{(k)} = \sum_{l=0}^{c(k)} b_{k,l}(\alpha)\lambda^l$ with $c(k) = k$ for RS theory and $c(k) = 2^k - 1$ for IPT. Abel's lemma does not apply to such expansions: $\psi = \lim_{k \to \infty} \psi^{(k)}$ can admit a singularity at some $\lambda = \lambda_*$, and nevertheless converge outside the disk with radius $|\lambda_*|$. For an illustration of this fact using a simple two-dimensional example see Appendix A.

**Epstein-Nesbet partitioning.** We have noted before that, given $H = F + gI$ where $F$ is a free theory (say a harmonic oscillator or a Fockian) and $g$ a physical coupling constant, setting $H_0 = F$ and $H_1 = gI$ is not necessarily the most judicious choice. Indeed another natural choice, Epstein-Nesbet (EN) partitioning [36], is often preferable. In the basis of eigenvectors of $F$, this is defined by

$$H_0 = \mathrm{diag}(H_{nn})_{n\geq 0} = F + g\,\mathrm{diag}(I_{nn})_{n\geq 0} \tag{6}$$

and $H_1 = H - H_0$. Perhaps because it is incompatible with the notion that perturbation theory should provide an expansion in powers of $g$, this choice is rarely used in the physical literature.

We now proceed to illustrate the virtues of combining EN partitioning with relaxed iteration through several examples. Although both RS and IPT give comparable results, in the following I focus on the latter (omitting subscripts) owing to its simpler structure and greater computational efficiency.

**Even anharmonic oscillators.** Even-order one-dimensional anharmonic oscillor $H = p^2 + x^2 + gx^{2s}$ are commonly cited to illustrate the difficulties encountered in RS theory [2, 9]. In the usual partitioning with $H_0 = p^2 + x^2$ and $\alpha = 1$, the RS coefficients for the ground state energy grow like $(-1)^n((s-1)n)!$ and the corresponding series diverges for any value of $g \neq 0$ due to branch point singularities accumulating at $g = 0$ [37, 38]. For $s = 2, 3$ the RS series for the ground state energy is Padé-summable, but for higher orders it is not; (generalized) Borel summation is however always possible [9]. It was noted in [39] that, using (6) but no relaxation, the RS series for the quartic oscillator $s = 2$ may converge for very small values of $g$. We checked that this is no longer true for $s = 3, 4$.

Relaxed perturbation theory with EN partitioning, on the other hand, always converges. Fig. 1 shows IPT energy iterates for the quartic oscillator at increasingly large coupling $g$; Fig. 2 in turn shows the relative errors for the quartic, sextic and octic oscillators with $g = 100$. Without restrictions, we find that $E^{(k)}$ converges to the exact eigenvalue $E$ as $k \to \infty$, albeit increasingly slowly as $s$ or $g$ gets larger. We emphasize that the convergence of $E^{(k)}$ is underpinned by the convergence of the eigenvector $\psi^{(k)}$ itself, as can be checked by computing the residuals $\|H\psi^{(k)} - \langle \psi_0, H\psi^{(k)}\rangle \psi^{(k)}\|$ (not shown).

**Herbst-Simon Hamiltonian.** The Herbst-Simon anharmonic oscillator with potential $V(x; g) = 2gx - 2gx^3 + g^2$ has a purely instantonic positive ground state energy $E \sim e^{-d/g^2}$ for some $d > 0$. This unusual non-analytic behaviour precludes a meaningful expansion in power of $g$, and shows that the standard RS expansion (identically zero in this case) can sometimes converge to the wrong value. As already noted, successful perturbative calculations of $E$ for g up to $\sqrt{0.3}$ have so far involved educated guesses and sophisticated resummation procedures [18, 21].

---

[4]IPT never does: even with $\alpha = 1$, $\psi_{\mathrm{IPT}}^{(k)}$ is a polynomial in $\lambda$ with $k$-dependent coefficients.

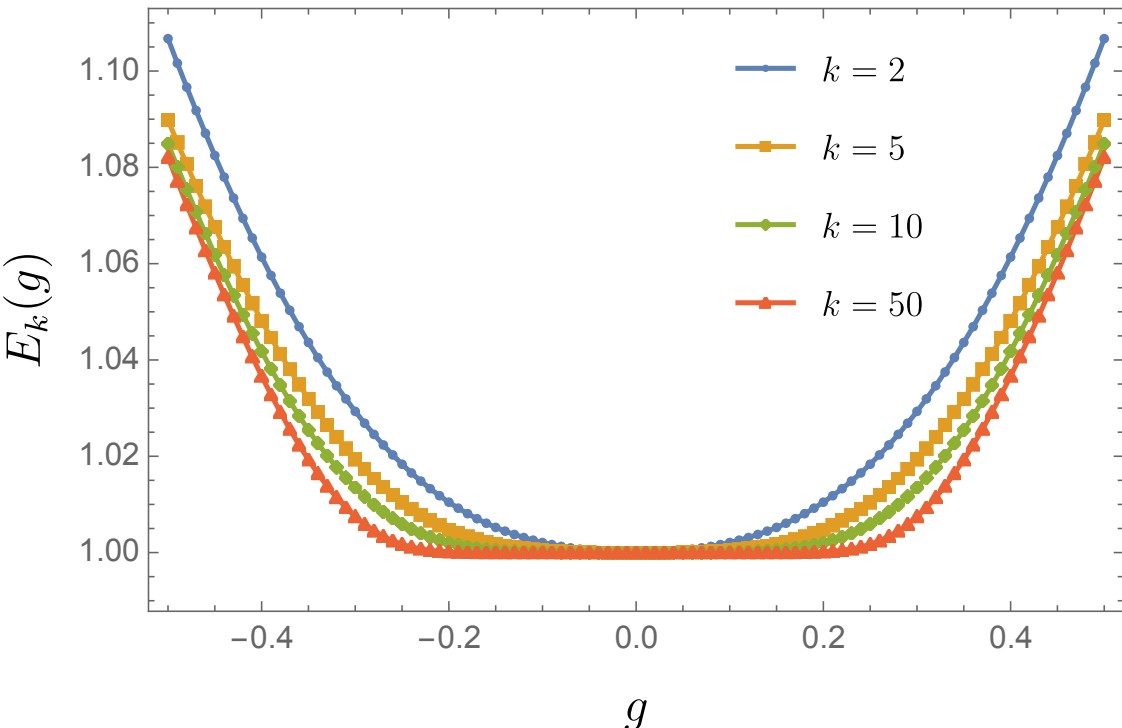

Figure 3: The Herbst-Simon ground state energy vanishes to all orders in $g$, but can be accurately computed using relatex perturbation theory, here IPT with $\alpha = 1/2$ and EN partitioning (6).

Using IPT with $\alpha = 1/2$ (again in EN partitioning) we easily compute $E$ for any $g$, to arbitrary accuracy. For instance, to obtain $E(\sqrt{0.3}) \simeq 1.11$ to two significant figures (the state of art using the methods of Ref. [21]), we only need a dozen iterations of IPT. With $k = 220$ iterations, we obtain all 13 figures of the exact result $[E(\sqrt{0.3}) - 1]/2 = 5.318357438655 \cdot 10^{-2}$ cited in [21, Table III]. Fig. 3 illustrates the convergence of $E^{(k)}$ to the exact value (indistinguishable from the red curve on this figure), with its non-polynomial behavior near $g = 0$.

**Zeeman problem.** We close with an example of historical as well as experimental significance, the hydrogen atom in a magnetic field, $H = -\Delta/2 - 1/r + g(x^2 + y^2)$ with $g = B^2/8$ (in atomic units). To compute Zeeman shifts it is useful to make use of the $SO(4, 2)$ dynamical symmetry of the problem [30, 40]. Thus, for the ground state $\psi$ with energy $E = -1/2 + \Delta E$, the Schrödinger equation may be reformulated as the generalized eigenvalue problem

$$(T_3 - 1 + gW)\psi = (\Delta E)S\psi \qquad (7)$$

where $T_3$ is a generators of a $\mathfrak{so}(2, 1)$ sub-algebra (with discrete spectrum), $W$ and $S$ can be expressed in terms of the other generators of $\mathfrak{so}(4, 2)$ [30, Chap. 9]. Using this representation with the partitioning $H_0 = T_3 - I$, it is possible to compute the RS coefficients of $\psi$ to large orders [3]. The resulting series diverges for any $g \neq 0$, but Padé approximants provide accurate estimates of $E$ up to $g \simeq 1$. In stronger fields, Borel resummation combined with order dependent mappings [41] or sequence transformations [42] give better results. These methods are all rather sophisticated and generally use additional information about the $g \to \infty$ limit, *e.g.* via a scaling of space which maps $g$ to the unit interval[5].

---

[5]The RS series for the Zeeman has been called "one of the most difficult summable divergent series encountered in physics" [42].

| Disorder $h$ | IPR | Iterations / CPU time ($s$) | | |
|:---:|:---:|:---:|:---:|:---:|
| | | IPT-AA | KS | LOBPCG |
| 1 | 0.004 | 396/19.4 | 8/2.1 | 85/4.4 |
| 2 | 0.11 | 64/3.2 | 7/1.9 | 65/3.4 |
| 5 | 0.68 | 21/1.0 | 7/1.8 | 62/3.3 |
| 10 | 0.92 | 14/0.64 | 5/1.3 | 45/2.4 |
| 50 | 0.99 | 7/0.30 | 4/1.1 | 32/1.7 |
| 100 | 0.999 | 5/0.22 | 3/0.93 | 27/1.4 |

Table 1: Exact (numerical) diagonalization of the ground state of the random-field Heisenberg spin chain. For disorder strengths $h \geq 3.7$ this system is many-body localized, implying that eigenstates are close to states in the computional basis [43] and display high inverse participation ratio (IPR). We used $L = 20$ spins, periodic boundary conditions and set the tolerance to $10^{-10}$. The Hamiltonian matrix was built using quimb [44], and Krylov-Schur (KS) and LOBPCG calculations were done with SLEPc [45]. For LOBPCG we used a trivial preconditioner, which proved more efficient than a diagonal preconditioner in this case. IPT was applied with Anderson acceleration (IPT-AA) with memory $M = 10$ [46]. Timings are with a 3.6 GHz 10-core Intel i9 processor.

A straightforward generalization of IPT to account for the $S$ matrix on the RHS of (7) reproduces these results without using *ad hoc* information , and without any limitation on $g$ (other than the slow convergence in very strong fields). For $B = 1$ (a strong field of $2 \cdot 10^5$ T) we obtain the correct result $E = -0.3312$ with just $k = 100$ iterations of IPT ($\alpha = 0.3$). For $B = 10$, $k = 1000$ iterations give $E^{(k)} = 3.72$, to be compared with $E = 3.25$ obtained in Ref. [42] using Weniger-type sequence transformations. Using elementary Aitken extrapolation this improves to $E^{(k)'} = 3.35$[6].

**IPT as eigensolver.** Like conventional RS theory, IPT is a scheme for computing analytical approximations of perturbed eigenvectors. However, in cases where $H_1$ is not block diagonal or can only be obtained in a finite basis set, as is common in quantum chemistry, the latter can also be used as an efficient numerical eigensolver. Given a matrix $H$, the simplest numerical implementation of IPT consists in replicating the relaxed perturbative calculations above: set $H_0 = \text{diag}(H)$, start from a basis vector, say $\psi_0 = (1, 0, \cdots, 0)$, and iterate $\psi^{(k+1)} = \alpha Q(\psi^{(k)}) + (1 - \alpha)\psi^{(k)}$ until a convergence condition is met. However, instead of using a fixed $\alpha$, it is more efficient to use an optimal convex combination of $F(\psi_{k-1})$ and previous iterates. This method, known as Anderson acceleration [47], constructs the new iterate as

$$\psi^{(k+1)} = \sum_{m=0}^{M} \beta_m Q(\psi^{(k-m)}), \tag{8}$$

where $M$ is a memory parameter and the positive coefficients $\beta_m$'s are recomputed at each step by minimizing the residual norm.

With this modification, IPT provides an eigensolver that can be remarkably efficient: a python implementation given in Appendix B proves up to 5x faster that state-of-the-art implementations of the Implicitly Restarted Lanczos and Locally Optimal Block Preconditioned Conjugate Gradient (LOBPCG) algorithms[7]. (Other recent methods, such as Generalized Davidson or Jacobi-Davidson, were slower in this case.) Table 1 reports the number of iterations and

---

[6]Aitken extrapolation of a sequence $s_n$ is $s'_n = (s_n s_{n+2} - s_{n+1}^2)/(s_n + s_{n+2} - 2s_{n+1})$.
[7]IPT can also be applied to non-Hermitian problems, in which case speed-up factors reach 100 or more [31].

CPU time required to compute the ground state of a random-field Heisenberg spin chain at various disorder strengths $h$, showing significant speed-ups in the many-body localized phase $h \geq 3.7$ (for notation and motivation see e.g. [48]). It is important to emphasize, however, that unlike Lanczos and other general-purpose eigenvalue algorithms, IPT has a limited applicability: due to its perturbative nature, the performance of IPT degrades rapidly when off-diagonal elements become large compared to the separation between its diagonal elements. For more details on the numerical aspects of IPT, I refer the reader to Ref. [31].

**Conclusion.**  "Relaxed perturbation theory" is the idea of applying relaxed iteration to compute the eigenvectors of perturbed operators, either using the well-known RS scheme, or, more conveniently, through the application of the non-linear operator $Q_{\text{IPT}}$. Combined with the Epstein-Nesbet partitioning prescription, this technique provides convergent approximations for the ground state wavefunctions (not just energies) of challenging Hamiltonians, including the purely instantonic Herbst-Simon Hamiltonian and the hydrogenic Zeeman problem, up to arbitrarily high field coupling strengths. My interpretation of these results is that "non-perturbative physics" is, in fact, squarely within the scope of perturbation theory.

**Acknowledgments.**   It is a pleasure to thank Maxim Kenmoe for stimulating my interest in perturbation theory, Anton Zadorin for a productive collaboration on the mathematical aspects of IPT, and an anonymous referee for constructive comments. Funding for this work was provided by the Alexander von Humboldt Foundation in the framework of the Sofja Kovalevskaja Award endowed by the German Federal Ministry of Education and Research.

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

# A    Relaxed pertubation theory: a toy example

To build intuition for the difference between relaxed RS and IPT theories on the one hand, and conventional perturbative expansions on the other hand, it is useful to consider the simple two-dimensional Hamiltonian

$$H = H_0 + H_1 \quad \text{with} \quad H_0 = \begin{pmatrix} 0 & 0 \\ 0 & 1 \end{pmatrix} \quad \text{and} \quad H_1 = \begin{pmatrix} 0 & g \\ g & 0 \end{pmatrix}. \tag{A1}$$

This matrix has ground state energy $E(g) = (1 - \sqrt{1+4g^2})/2$. At the exceptional points $g = \pm i/2$, this function has branch-point singularities which signal that $H$ is no longer diagonalizable. Consequently, the conventional RS expansion of $E(g)$ can only converge in a centered disk $D_{1/2}$ with radius $1/2$, although $E(g)$ is analytic everywhere except at $g = \pm i/2$. Let us now examine how relaxed perturbation theory behaves in this case.

**Relaxed IPT.**    The simpler case to analyze is IPT. For the ground state we have

$$\psi_{\text{IPT}}^{(k)} = \begin{pmatrix} 1 \\ q_\alpha^k(0) \end{pmatrix} \quad \text{with} \quad q_\alpha(x) = \alpha g(x^2 - 1) + (1-\alpha)x. \tag{A2}$$

Here the superscript denotes $k$-fold composition. Evaluated at its fixed point $x_* = (1-\sqrt{1+4g^2})/2g$, the Jacobian of $q_\alpha$ reads $q_\alpha'(x_*) = 1 - \alpha\sqrt{1+4g^2}$. This quantity has modulus smaller than 1 inside a cardioid-shape domain in the complex plane, which is therefore the domain of convergence of IPT. This domain excludes the singular points $\pm i/2$, but extends along the real axis up to arbitrarily large values of $g$ for sufficiently small $\alpha$. For $\alpha = 1$ (no relaxation), this domain is strictly greater than the disk $D_{1/2}$.

**Relaxed RS theory.**    The RS expansion of the ground state with $\alpha$-relaxation to $k$-th order reads [35]

$$\psi_{\text{RS}}^{(k)} = \sum_{l=0}^{k} \sum_{s=1}^{l} \binom{l-1}{s-1} \alpha^s (1-\alpha)^{l-s} g^s a_s \tag{A3}$$

where $a_s$ are the coefficients of the standard (unrelaxed) RS expansion of $\psi$. This expression is not a power series in $g$, hence its domain of convergence is not restricted to the disk $D_{1/2}$.

Fig. A1 illustrates these constructions graphically. From this figure it is apparent that, at least in this simple case, IPT at a given order $k = 10$ approximates $E(g)$ better than RS theory at the same order, and that in both cases $\alpha = 1/2$ provides extended convergence with respect to the $\alpha = 1$.

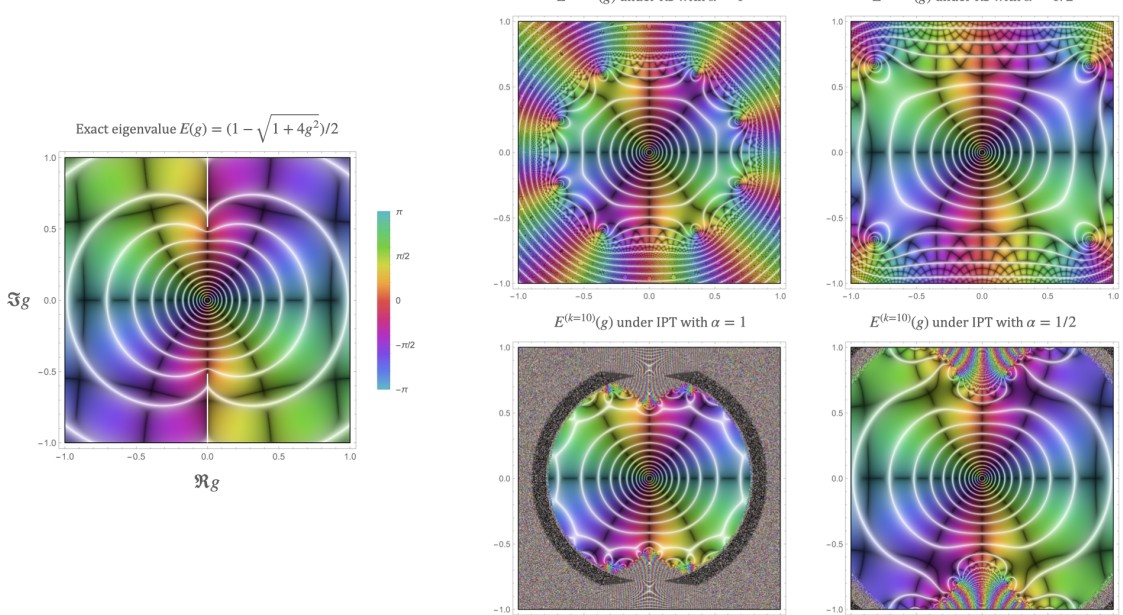

Figure A1: Different ways to construct a complex function with singularities with polynomials. In all plots color indicates argument, brightness indicates log-modulus, and saturation indicates real and imaginary magnitude. Left: Left: The exact eigenvalue $E(g) = (1 - \sqrt{1 + 4g^2})/2$, with its branch-point singularities at $g = \pm i/2$. Its Taylor expansion in $g$ must diverge outside the disk $D_{1/2}$ with radius $1/2$. Right, top row: the conventional RS series ($\alpha = 1$) is valid inside a circle with radius $1/2$, but the same expansion with $\alpha = 1/2$ has a larger, non-circular domain of convergence. Right, bottom row: IPT with $\alpha = 1$ converges inside a cardioid shaped domain which is stricly larger than $D_{1/2}$; with $\alpha = 1/2$ this domain extends yet further.

## B   Python code for IPT-AA

```python
import aa # Anderson acceleration from https://github.com/cvxgrp/aa

import numpy as np
import time

def eigs_ipt(H, i = 0, v0 = None, mem = 10, maxiter = 1000, tol = 1e-
                                    10):

    dim = H.shape[0]
    H0 = H.diagonal()   # Epstein-Nesbet partitioning
    g = H0[i] - H0; g[i]=1; R0 = 1/g; R0[i] = 0   # Reduced resolvent
    aa_wrk = aa.AndersonAccelerator(dim, mem)     # Initialize
                                    accelerator
    e = np.zeros(dim); e[i] = 1   # Unperturbed eigenvector (basis
                                    state)
    if v0 is not None:
        w = v0
    else:
        w = e

    def Q(v, R0, H0):      # Quadratic operator, eq. (3) in main text
        H1v = H @ v - np.multiply(H0, v)
        return(e + np.multiply(R0, H1v - H1v[i]*v), H1v)

    err = [1]
    l = 0

    tic = time.time()
    while l <= maxiter and err[-1]>tol:
        v = w
        w, H1v = Q(v, R0, H0)   # Iterate Q
        l += 1
        aa_wrk.apply(w, v) # Anderson acceleration
        E = H0[i] + H1v[i] # Evaluate energy
        err.append(np.linalg.norm(v - w)) # Estimate error
    toc = time.time()

    print('Iterations:', l - 1)
    print('Time:', toc - tic)

    return(E, v)
```