# Peer review of "Perturbation theory without power series: iterative construction of non-analytic operator spectra"

_SciPost Physics_

## Round 1 · Referee Report · Anonymous (Referee 1) · 2022-3-14

Report

This paper discusses techniques for numerically solving the quantum-mechanical spectral problem. One standard approach is to compute RS perturbative series and try to resum it if it is divergent which is often the case. Alternative approaches discussed here include iterative perturbation theory IPT (previously also discussed in the author’s paper [31]). The current paper focuses on relaxation and EN partitioning defined in Eqs. (4) and (6). The author introduces these methods and demonstrates their strength on several explicit examples: quartic, sextic, octic anharmonic oscillators with strong quartic coupling, Herbst-Simons Hamiltonian (whose formula on p.5 did not fit in the page) and Zeeman problem.

He then shows that IPT ideas could be used to speed up exact diagonalization algorithms for the random-field Heisenberg spin chain with 20 sites. This part of the paper stands a bit apart from the rest.

I have mixed feelings about this paper. On the one hand the results are curious, on the other hand no important problems have been solved with the new method. I have a feeling that all QM examples the author considered could be very well solved with alternative techniques (like variational methods in a space of moderate dimension). There is no demonstration that the speed up provided by the new ideas is crucial for the advancement of physics. The claimed advances are on the conceptual side, but they rest empirical. It is mentioned that the new methods cannot be shown to diverge based on Abel’s lemma. But the absence of divergence does not imply convergence. No real explanation is given for why the new methods are supposed to converge, as it seems so empirically. For IPN, a convergence result for finite-norm perturbations from [31] is cited, but the considered perturbations do not have finite operator norm, so that result does not apply.

The paper is also very terse. This might be due to the fact that, judged by the format of the original submission, this paper was originally prepared for PRL. PRL has page limits but SciPost does not. It would be great if the authors could somewhat expand the paper to make it generally less terse, as well as address the points mentioned in the previous paragraph, if they have anything interesting to say about them.

There are also misprints easily catchable by a spellchecker and regrettable formatting mistakes like equations which don’t fit the page.

---

## Round 1 · Referee Report · Anonymous (Referee 3) · 2022-3-18

Report

This paper discusses efficient strategies for using iterative perturbation theory (IPT) and argues that IPT can be used way beyond the scope of simple Rayleigh Schrödinger perturbation theory which may yield divergent power series. I find this paper very well written, providing a thorough background on the problem of divergent perturbation series and traditional techniques to deal with them. It is argued that resummation strategies are often not very useful in practice and iterative perturbation theory is offered as an alternative, in particular if it is combined with a relaxation technique, where the updating scheme in each iteration also uses information of previous iterations.

The points made in this work are underlined by numerical results in several examples, which yield divergent series in traditional perturbative approaches and very good agreement with exact solutions of these models is achieved. Furthermore, the performance of the scheme is shown in comparison with other high performance eigensolvers for finding the ground state of the random field Heisenberg model.

I find this paper very well written, with a great motivation for the problem and the results are certainly interesting. My main criticism is that it seems that all the techniques used in this work are known for a long time (as shown by the references), and it is not sufficiently clear what is the original contribution of this paper. It seems that even the employed relaxation strategy has been introduced before, but perhaps not explicitly applied to the examples provided here? I tend to agree with the first referee that the problems shown as examples do not seem to be that difficult compared to the hardest many-body problems found e.g. in frustrated systems. Is it possible to apply these techniques for example to frustrated magnets, or the Hubbard model at finite doping? It would be great to see the performance of the technique in such 'real world' scenarios.
As also touched upon in the main text, the random field Heisenberg model at very strong disorder is not an extremely hard problem in that sense, since its eigenstates very deep in the MBL phase have a simple structure. This being said, I am a bit surprised that the fidelity at h=100 is "only" 0.999. Does this perhaps imply that the subtle structure, such as many-body resonances are not captured well by this perturbation theory?

In summary, I think this paper is interesting and if the original contributions can be highlighted convincingly I would recommend it for publication in SciPost Physics Core. I agree with the first referee that the text would benefit from expanding certain parts. Perhaps adding a section explaining the practical use of the algorithm would be helpful (something like a plain english version of the attached python code).

Minor comments:

+ It seems that there is a typo in Eq. (1)
+ Would it be possible to add a (perhaps black dashed) line on top of the data shown in Fig. 3 to represent the exact result for comparison? The text says the data is indistinguishable from the exact data, but it would be nice to see this explicitly.
+ The email address of the author appears to be not set.

---

## Round 1 · Referee Report · Anonymous (Referee 2) · 2022-3-18

Strengths

The work presents a new combination of approaches which could lead to interesting future applications.

Weaknesses

1) The approach is a new combination of previous methods, hence not theoretically novel. Its general practical utility is also not clear based on the examples provided.

2) Given that the convergence properties cannot be assessed analytically, their numerical characterization is incomplete.

3) The manuscript is unnecessarily less easy to read due to being too succinct.

Report

The tendency of quantum perturbation theory to produce a divergent series is a well-known problem, with no good general solution to date. The manuscript presents a contribution to this field, combining Iterative Perturbation Theory presented with mathematical rigor in a recent study by the Author and collaborators [31], together with relaxation and Epstein-Nesbet splitting. It is shown numerically that this combination provides convergent schemes for accurate solutions of several single-particle problems with known pathologies in their perturbation series, as well as for disordered 1D chains. However, unfortunately I currently cannot recommend its publication in SciPost Physics due to the following issues:

1) While interesting, given the fact that the manuscript does not present any new approach, only a new combination of previously-known ones, its novelty is somewhat limited from the "theoretical" perspective. And from a "practical" perspective the utility of the new combination of methods is also not clear - the quantum mechanical problems can probably be approached by other methods for solving ODEs, not related to perturbation theory. As for the many body example, the new combination of approaches seems to be advantageous only in the many-body localized regime, which is amenable to tensor network techniques (due to the low entanglement) or strong disorder renormalization group approaches. For making a better case for the new combination of approaches, a study of 2D disordered systems could help. In addition, to give advantage in the delocalized phase, it might be useful to try to start the iterations there from the clean system limit (perhaps also in the noninteracting limit in terms of Jordan-Wigner fermions).

2) As mentioned above, it appears that the Author cannot provide analytical proofs and conditions for convergence of the method for any of the discussed systems, and therefore resorts to numerical demonstrations. However, even those do not appear to be systematic enough. (a) For example, although the convergence seems to feature a power-law dependence on the number of iterations, no attempt is made for characterizing its dependence on the problem parameters. (b) Furthermore, for the relaxation parameter \alpha only the values 0 and 1/2 are used. A systematic numerical characterization of the dependence of the convergence properties on the value of \alpha would be much desirable. (c) Similar data regarding the convergence properties of the eigenstates should be provided (instead of just mentioning that it can be done). (d) In the many body example, it would be interesting to characterize the performance of the method also for excited states, and thus understand its dependence on the energy density.

3) The manuscript is written in a very succinct way with many details given only figure captions, or omitted altogether relying on references on on the reader to figure them out (some representative examples: for the Zeeman system the Hamiltonian is given after subtracting the part linear in the angular momentum along the magnetic field direction without mentioning this step; The Hamiltonian of the disordered Heisenberg model is not specified explicitly; no reference is given on, e.g., Aitken's method or Wigner's 2n+1 theorem). This might be appropriate for a short paper format but not for SciPost Physics, and makes the manuscript unnecessarily harder to follow.

3) There seems to be typos, even in some equations:
In the unnumbered displayed equation before equation (1), it seems that \lambda should only multiply H_1 and not the entire second term on the r.h.s. This issue seems to persist in the inline equation defining the choice of \mathcal{E} leading to equation (1) and in equation (1) itself, as well as in the inline equation following equation (6).
Also, 3 lines after equation (2), "final" should probably be "finite".

Requested changes

1) Demonstrating more use cases for the presented approach.

2) Providing a much more extensive numerical convergence characterization.

3) Making the manuscript more "reader-friendly" given that there is no length limit.

4) Correcting typos.

---

## Editorial Decision

awaiting_resubmission